# Trends in prevalence and correlates of tobacco use among school-going adolescents in Bhutan: A secondary data analysis of the 2004–2019 Global Youth Tobacco Survey

Tshewang Gyeltshen[1,2,3] *, Mahbubur Rahman[1]

1 Graduate School of Public Health, St. Luke's International University, Tokyo, Japan, 2 Division of Population Data Science, National Cancer Centre, Institute for Cancer Control, Tsukiji, Tokyo Japan, 3 Tsirang Hospital, Ministry of Health, Royal Government of Bhutan, Thimphu, Bhutan

* tshewangg@jdwnrh.gov.bt

**Data Availability Statement:** The data relevant to this study are available from https://extranet.who.int/ncdsmicrodata/index.php/catalog/GYTS.

## Abstract

Tobacco use remains a major public health challenge globally. In Bhutan, despite the implementation of strict tobacco control measures, the tobacco uses among the students continue to be alarmingly high compared to neighboring countries. This study aims to analyze the trends and correlates of tobacco use among students in Bhutan, utilizing the nationally representative Global Youth Tobacco Survey (GYTS) data from multiple survey years. Secondary analyses of GYTS data collected during 2004–2019 with 12,594 students aged 11–18 years were employed. Use of tobacco was defined as either smoked or smokeless tobacco use in last 30 days of the survey. Prevalence was estimated over time and multivariable log-binomial regression was used to determine the correlates of current tobacco use. Overall tobacco use prevalence increased from 18.5% in 2004 to 27.3% in 2019. Males had higher prevalence (20.4% in 2019) than females (7.0% in 2019). Smokeless tobacco use increased substantially from 8.2% to 19.4% over the study period. Earlier age of initiation had adjusted odds ratio (aOR) of 9.2 for <11 years and 12.8 for 13–16 years vs. never smoking), betel quid use (aOR 3.3), peer pressure (aOR 3.6), and lower cost were significant correlates of adolescent tobacco use. Despite tobacco control policies, tobacco use among Bhutanese students is high and has been increasing over time, especially smokeless forms. Tobacco uses regulation, targeted interventions for high-risk junior high school students, and addressing social influences are urgently needed to curb this epidemic. Sustained tobacco use surveillance and public health action is imperative to protect students in Bhutan from this harmful habit.

## Introduction

Tobacco use is the leading cause of preventable death worldwide. Globally, about 7 million deaths are attributed to tobacco use annually [1]. The harmful use of tobacco has been linked to various health problems such as cardiovascular disease, cancer, and respiratory diseases.

**Funding:** The authors received no specific funding for this work.

**Competing interests:** The authors have declared that no competing interests exist.

Tobacco use among youth is particularly concerning, as it can lead to nicotine dependence and long-term health problems. An estimated 1.2 billion people aged 15 years and above are projected to be tobacco users by the year 20230 based on current global tobacco use prevalence [2].

Prevalence of tobacco use among youth has been increasing across the world, and Bhutan is no different. The Global Youth Tobacco Survey 2019 reported that 22.2% of the school going students were current tobacco users [3]. Another area of problem in Bhutan's tobacco control measure is use of betel quid with dried tobacco. An alarming 48.9% of the Bhutanese students are found to be current user of the betel quid [3].

The royal government of Bhutan has enforced various rules and regulations to curb the tobacco use. Tobacco use in Bhutan was viewed as reprehensible from both religious and health perspectives, which led to its endorsement for a ban during the 70th National Assembly Session of pre-democracy Bhutan [4] in 1991. Bhutan ratified the World Health Organization Framework Convention on Tobacco Control (WHO-FCTC) in the year 2004 followed by the nationwide ban. In June, 2010 the first democratically elected government of Bhutan enacted the Tobacco Control Act, which was followed by the enforcement of a comprehensive nationwide ban on tobacco use in the same year [5]. Amendments to the bill saw relaxation of the regulations allowing limited quantities for personal consumptions in the year 2014 [6]. The law not only prohibited the illegal sale of tobacco but also the use of tobacco products in public places, with penalties specified under the Bhutan Penal Code Act of 2004 [7].

Although Bhutan has a strong legal and regulatory framework for tobacco control, the prevalence of tobacco uses among Bhutanese students has remained as a significant concern. The problem is exacerbated by widespread smuggling and a thriving black market for tobacco within the country. Despite efforts, the prevalence of tobacco uses among students in Bhutan have increased. The Global School Based Student Health survey in 2016 reported prevalence of 29% tobacco use among the students [8].

This study aims to examine the trends in prevalence and correlates of tobacco use among students in Bhutan, utilizing data from multiple points of the Global Youth Tobacco Surveys (GYTS).

## Materials and methods

### Study design

This is a cross-sectional study employing secondary data analysis of nationally representative Global Youth Tobacco Survey across five years: 2004, 2006, 2009, 2013, and 2019 conducted in Bhutan.

### Setting

Bhutan is a Himalayan Kingdom located in the eastern flank of the Himalayas, sandwiched between India and China. With the population of 0.75 million Bhutan is known for its pristine environment and rich cultural heritage. It is administratively divided into 20 districts with Thimphu as the largest city and the capital. Bhutan, internationally recognized for its Gross National Happiness development index, prioritizes health and wellbeing as central to its developmental goals, ensuring free access to health services in both traditional and modern allopathic medicine [9].

### Study population

Bhutanese school going adolescents who were aged 11–18 years in the years of 2004, 2007, 2009, 2013, and 2019 were included for the purpose of this study. While the GYTS survey was

designed to target students aged 13–15 years, the available data includes ages ranging from 11 years or younger to 18 years or older. Therefore, we utilized this broader age range for our analysis. A total of 12594 respondents were included for the purpose of descriptive analysis and 4712 respondents from the 2019 survey wave were used in multivariable analysis to examine the correlates of tobacco use.

## Patient and public involvement

We used the secondary data from the GYTS surveys conducted by the Ministry of Health, Royal Government of Bhutan, across five survey years: 2004, 2006, 2009, 2013, and 2019. While public may have been involved in primary survey result dissemination, they were not involved in the current analysis.

## Ethics statement

Ethical approval is not necessary for this study, as it is a secondary data analysis of surveys previously conducted by the Ministry of Health, Royal Government of Bhutan. The Research and Ethics Board of Health, Royal Government of Bhutan, has provided primary ethical clearance for all the surveys in each year.

## Data source, sampling, and sample size

Global Youth Tobacco Survey conducted in 2004, 2007, 2009, 2013 and 2019 in Bhutan was used for the purpose of this analysis. All the datasets are available as public use files in WHO microdata repository (https://extranet.who.int/ncdsmicrodata/index.php/catalog/GYTS). The outcome variables for the study were the tobacco use derived as composite variable across all the survey years as "ever used tobacco", "current smokers" and "current smokeless tobacco users". To derive the "ever used tobacco" three questions; *"Have you ever tried or experimented with cigarette smoking", "Have you ever tried or experimented with any form of smoked tobacco products other than cigarettes"*, and *"Have you ever tried or experimented with any form of smokeless tobacco products"* were used. "Current smokers" were defined by the questions *"During the past 30 days, on how many days did you smoke cigarettes?"* and "*During the past 30 days, did you use any form of smoked tobacco products other than cigarettes*". "Current smokeless tobacco users" is defined by the question "*During the past 30 days, on how many days did you use smokeless tobacco*?". E-cigarette use was not included in any of the questionnaires and, therefore, was not considered in this analysis.

 The Global Youth Tobacco Survey (GYTS) is a global school-based tobacco use survey spearheaded by the World Health Organization (WHO) and the Centers for Disease Control and Prevention (CDC). It serves as a key tool for monitoring tobacco use among youth and guiding countries towards tobacco control measures in accordance with WHO Framework Convention on Tobacco Control (FCTC). Utilizing a two-stage cluster sampling method, the survey targets schools, and students within specific grades, ensuring a probability-proportional selection based on enrollment numbers. Participants anonymously answer a standardized set of 54 questions in either English or local languages, covering various aspects of tobacco use, including prevalence, accessibility, cessation interest, and exposure to secondhand smoke and advertising. The methods and more detailed information about the GYTS are available on the CDC website [10].

## Statistical analysis

Once the dataset was retrieved from the WHO public use files repository, data from all the survey waves were merged and observations requiring change of values and value labels were

managed according to the codebook provided with the dataset. The acquired data were managed and analyzed using the open software R version 4.3.1, (R Core Team (2023). R Foundation for Statistical Computing, Vienna, Austria). We utilized various R packages for data management and subsequent statistical analyses as provided in the S1 Data. The demographic characteristics of participants were described as frequencies; mean (SD) or median (interquartile range) and percentages as appropriate. To account for the complex survey design of the GYTS, we used the survey package in R. Using this package, we created a survey design object with the following specifications: primary sampling units (PSUs) were specified using the ids argument, stratification was incorporated using the strata argument, and the final survey weights were applied using the weights argument. This approach allowed us to properly account for the nested structure of the data and ensure accurate variance estimation. Annual Average Prevalence Change (AAPC) between 2004 and 2019, was calculated using a linear regression model that assumed exponential change in prevalence over the survey years. This model relates prevalence data points after the baseline year ($t_0$) to the baseline prevalence ($Y_o$) through the equation: $Y_{t_i} = Y_o(1 - b)^{(t_i - t_0)}, \rightarrow ln(Y_{ti}) = ln(Y_o) + (t_i - t_0) * ln(1 - b\%)$. AAPC ($b\%$) was calculated using $\beta$ from a linear regression of $ln(Y_i)$ against $t_i$, and the formula $AAPC = 1 - EXP(\beta)$, as described in UNICEF technical note to calculate Average Annual Reduction Rate for underweight prevalence [11]. All the surveys included in the study are national surveys considered nationally representative in Bhutan.

Log-binomial multivariable logistic regression models were fitted to assess the correlates of tobacco use prevalence. The outcome was current tobacco users derived from composites of variables "current smokers" and "current smokeless tobacco users". To address the issue of data heterogeneity in repeated cross-sectional surveys and to accurately reflect the current situation, only data from the 2019 GYTS was used for covariate analysis. The covariates used in the analysis include age of exposure, current use of betel quid, peer pressure, cost of tobacco, best friend's smoking status, second-hand smoking, and media promotion of tobacco among others as described in Table 3. Both crude and adjusted effects were estimated for the non-missing sample. P-value of <0.05 was considered statistically significant. Variables were included in the model selection process based on authors' knowledge of the literature and available information in the survey datasets. The step function in R was used to select the final model after defining the null and full model specifying forward selection procedure. This function ranks according to models' goodness of fit using a stepwise forward Akaike information criterion (AIC) method.

## Results

The characteristics of the sampled respondents are summarized in Table 1 based on all survey years separately. Due to data heterogeneity, only those characteristics which are common across all the survey years were summarized. Majority of the survey participants were from the age group 13–16 years old with relatively equal distribution of sex across all the survey years. Most students reported having no pocket money, though the percentage with over 500 Ngultrum increased from 1.9% in 2004 to 6.3% in 2019. Exposure to anti-tobacco messages through media and events increased over time, from 18.6% and 59.1% in 2004 to 38.7% and 70.8% in 2019. The percentage of participants who saw tobacco use on TV/movies was higher in later years (44.2% vs.70.8%). Among current smokeless tobacco users, the frequency of participants "who thought it is difficult to quit" increased from 59.1% in 2004 to 62.1% in 2019. Similarly, those who wanted to quit also increased from 11.4% to 20.3%. Exposure to secondhand smoke at home decreased substantially from 31.7% in 2004 to 19.2% in 2019.

**Table 1. Characteristics of adolescent survey participants in Global Youth Tobacco Survey, Bhutan, 2004–2019.**

| | 2004 (N = 1807) | | 2006 (N = 1921) | | 2009 (N = 1835) | | 2013 (N = 2319) | | 2019 (N = 4712) | |
|---|---|---|---|---|---|---|---|---|---|---|
| | n | Weighted % | n | Weighted % | n | Weighted % | n | Weighted % | n | Weighted % |
| **Age of the respondents** | | | | | | | | | | |
| 12 years or younger | 204 | 11.45 | 274 | 15.27 | 217 | 11.49 | 266 | 12.12 | 226 | 5.54 |
| 13–15 years old | 870 | 46.78 | 1130 | 59.91 | 1019 | 54.45 | 1378 | 60.5 | 2344 | 52.62 |
| 16 years old or older | 700 | 41.77 | 476 | 24.83 | 564 | 34.06 | 658 | 27.38 | 2133 | 41.85 |
| **Sex of the respondents** | | | | | | | | | | |
| Male | 820 | 51.31 | 889 | 49.82 | 794 | 49.78 | 1017 | 47.52 | 2157 | 52.55 |
| Female | 921 | 48.69 | 950 | 50.18 | 985 | 50.22 | 1293 | 52.48 | 2532 | 47.45 |
| **Best friend's Smoking** | 710 | 40.36 | 791 | 41.75 | 804 | 45.97 | - | - | 2448 | 52.55 |
| **Parent's smoking** | 333 | 18.36 | 362 | 19.08 | 302 | 16.59 | - | - | 999 | 20.86 |
| **Secondhand Smoking at home** | 575 | 31.69 | 627 | 32.76 | 587 | 32.4 | 363 | 15.88 | 885 | 19.22 |
| **Amount of pocket money** | | | | | | | | | | |
| No pocket money | 1353 | 76.43 | 1323 | 69.84 | 1197 | 63.6 | 492 | 21.2 | 932 | 20.12 |
| Less than Nu. 500 | 389 | 21.74 | 542 | 29.03 | 559 | 32.06 | 1701 | 73.53 | 3469 | 73.58 |
| More than Nu.500 | 32 | 1.85 | 21 | 1.13 | 73 | 4.34 | 122 | 5.27 | 302 | 6.3 |
| **Anti-tobacco media messages** | 746 | 43.32 | 918 | 49.21 | 985 | 54.84 | 1694 | 73.83 | 3151 | 67.58 |
| **Anti-tobacco Event messages** | 316 | 18.57 | 430 | 23.63 | 352 | 19.8 | 759 | 32.92 | 1797 | 38.65 |
| **Saw tobacco use in Tv/Movies** | 0 | | 830 | 44.16 | 0 | | 1573 | 68.51 | 3297 | 70.77 |
| **Advertisement in print media** | 1204 | 70.28 | 1382 | 73.77 | 1315 | 74.11 | - | - | - | - |
| **Advertisement through events** | 812 | 58.39 | 797 | 58.01 | 766 | 50.78 | - | - | - | - |
| **Comfortable in Social Gathering with Smoking** | 315 | 18 | 383 | 20.61 | 409 | 22.98 | 321 | 14.17 | 771 | 16.69 |
| **Have products with tobacco logo** | 187 | 11.18 | 211 | 11.57 | 291 | 16.97 | 322 | 14.67 | 529 | 11.93 |
| **Danger of tobacco use taught in last year** | 1010 | 58.29 | 1098 | 58.25 | 1100 | 61.62 | 1360 | 58.56 | 3311 | 69.94 |
| **Anti-tobacco messages in Media in last 30 days** | 746 | 43.32 | 918 | 49.21 | 985 | 54.84 | 1694 | 73.83 | 3151 | 67.58 |
| **Anti-tobacco messages in Events** | 316 | 18.57 | 430 | 23.63 | 352 | 19.8 | 759 | 32.92 | 1797 | 38.65 |
| **Think it is difficult to quit after using smokeless tobacco** | 1009 | 59.07 | 992 | 52.91 | 1093 | 60.56 | 1472 | 64.45 | 2943 | 62.14 |
| **Want to quit smokeless tobacco** | 204 | 11.36 | - | - | 211 | 12.06 | 357 | 15.53 | 913 | 20.27 |

Table 2 presents overall and sex-stratified weighted prevalence of tobacco use among Bhutanese students across the 5 survey years from 2004 to 2019. The prevalence of "ever used tobacco" and "current use" (both smoked and smokeless), "smoked", and "smokeless" tobacco use are presented. Among males, "ever tobacco users" increased from 22.4% in 2004 to 32.4% in 2019. Current tobacco users (smoked and smokeless) also increased from 13.3% to 20.4%. Smoking prevalence in males increased from 10.8% to 15.7% and smokeless tobacco use increased from 5.9% to 14.9% over the survey years. Female respondents had lower prevalence of ever tobacco use throughout all the survey years but increased slightly from 13.1% in 2004 to 14.9% in 2019. Current tobacco uses also increased slightly from 4.9% to 7.0% during the same period. Smoking prevalence in females remained relatively steady around 3.5–4.7% across all the survey years. Smokeless tobacco use increased slightly: from 2.1% in 2004 to 4.5% in 2019.

Fig 1, shows prevalence trend over the survey years by sex. Overall, females had lower prevalence across all the four categories of tobacco use: ever users of tobacco, current users, smokers, and smokeless tobacco user. The average annual change for the ever-used tobacco was 2.1%, while the change for current tobacco user was 2.4%. In contrast, the AAPC for current smokeless tobacco use was significantly higher, at 5.8%, indicating more than a two-fold increase compared to the 2.0% change for current smokers over the survey years.

**Table 2. Overall and sex stratified weighted prevalence of tobacco use during the survey years.** **\*\*sigf-)\*\* APPC Change.**

|  | 2004 (N = 1807) | 2006 (N = 1921) | 2009 (N = 1835) | 2013 (N = 2319) | 2019 (N = 4712) | AAPC |
|---|---|---|---|---|---|---|
| **Ever used (smoked/smokeless)** | **36.05** | **37.73** | **41.13** | **47.81** | **47.21** | **2.12** |
| *Male* | 22.43 | 25.24 | 27.06 | 27.49 | 32.42 | |
| *Female* | 13.14 | 12.07 | 13.68 | 20.36 | 14.86 | |
| **Tobacco users (smoked/smokeless)** | **18.55** | **21.03** | **24.22** | **24.35** | **27.33** | **2.41** |
| *Male* | 13.29 | 15.20 | 17.83 | 16.77 | 20.35 | |
| *Female* | 4.85 | 5.46 | 6.00 | 7.59 | 6.98 | |
| **Smokers** | **14.69** | **15.61** | **18.67** | **16.79** | **20.28** | **2.03** |
| *Male* | 10.76 | 11.40 | 14.06 | 12.25 | 15.65 | |
| *Female* | 3.46 | 3.82 | 4.36 | 4.56 | 4.65 | |
| **Smokeless Tobacco Users** | **8.22** | **11.58** | **13.1** | **20.04** | **19.39** | **5.77** |
| *Male* | 5.87 | 8.14 | 9.79 | 13.51 | 14.89 | |
| *Female* | 2.12 | 3.11 | 2.95 | 6.52 | 4.47 | |

## Correlates

To identify the correlates of tobacco use, the log-binomial multivariable regression analysis was conducted. Table 3 shows the unadjusted univariate analysis of the identified variables and the adjusted odds ratio (aOR) of the best fit model. The univariate analysis identified key variables which were statistically significant. The variables which were statistically significant in univariate analysis were further adjusted for other variables in the best fit model and final best fit model was selected using step function in R. Our analysis revealed key variables such as "age of initiation", "peer pressure", "student age", "current users of betel quid (DOMA)",

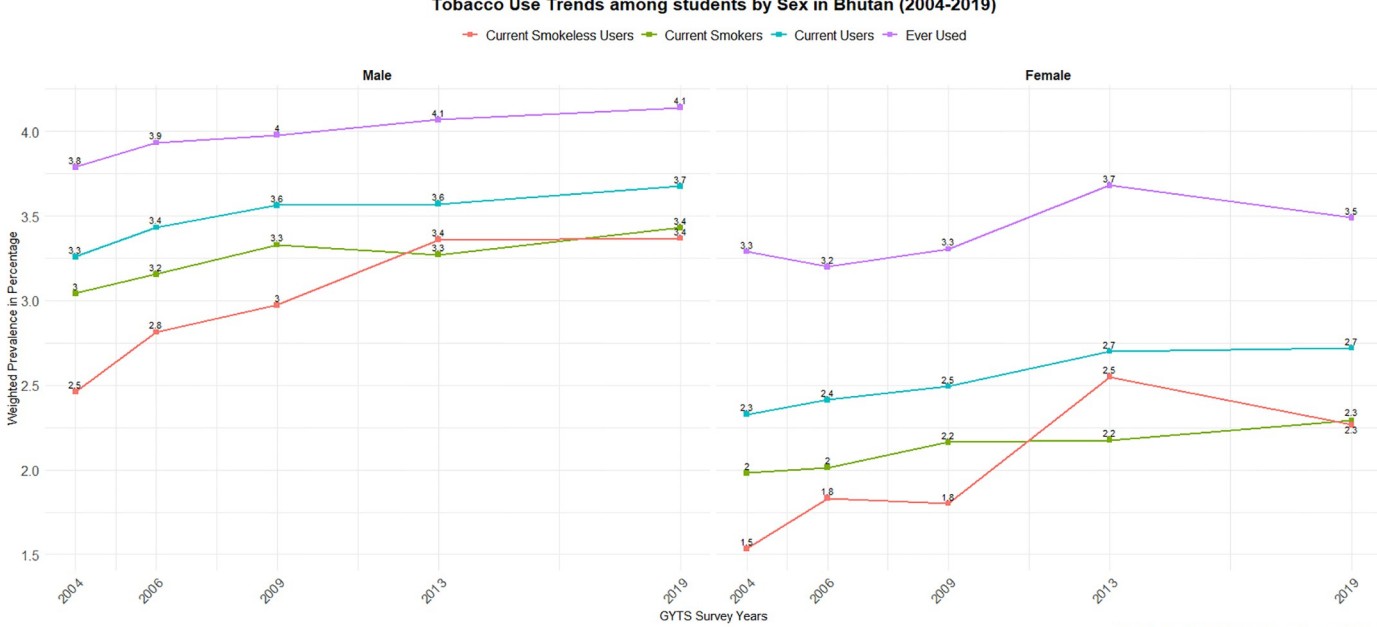

**Fig 1. Sex stratified tobacco use prevalence of students in Bhutan, 2004–2019.**

**Table 3. Log binomial multiple regression output, GYTS 2019.**

| Characteristics | Univariate Analysis | | | Adjusted best fit model | | |
|---|---|---|---|---|---|---|
| | OR | 95% CI | p-value | aOR | 95% CI | p-value |
| **Age of Exposure** | | | | | | |
| Never tried smoking cigarette | Ref | - | - | Ref | - | - |
| 11 years old or younger | 30.6 | 21.8–35.2 | <0.001 | 9.2 | 6.2–13.7 | <0.001 |
| 13–16 years old | 42.4 | 43.1–58.5 | <0.001 | 12.8 | 9.6–17.1 | <0.001 |
| **Current Users of DOMA (ref: NO)** | 5.2 | 4.2–6.5 | <0.001 | 3.3 | 2.5–4.2 | <0.001 |
| **Peer Pressure (ref: NO)** | 9.0 | 7.0–11.6 | <0.001 | 3.6 | 2.6–4.8 | <0.001 |
| **Tried other smoking products (ref: NO)** | 14.3 | 11.6–17.5 | <0.001 | 2.7 | 2.1–3.5 | <0.001 |
| **Cost per 10 packets of cigarette** | | | | | | |
| I don't know | Ref | - | - | Ref | - | - |
| Paid less than Nu. 250 | 8.6 | 7.3–10.2 | <0.001 | 1.7 | 1.3–2.3 | <0.001 |
| Paid more than Nu. 250 | 6.9 | 4.4–11.1 | <0.001 | 1.2 | 0.6–2.2 | 0.6 |
| **Best Friend's Smoking Status (ref: NO)** | 7.6 | 6.2–9.3 | <0.001 | 1.8 | 1.4–2.4 | <0.001 |
| **Able to buy near school (ref: NO)** | 97.0 | 0.9–1.1 | 0.6 | 0.8 | 0.6–1.1 | 0.13 |
| **Secondhand Smoking at home (ref: NO)** | 3.2 | 2.6–4.0 | <0.001 | 1.8 | 1.4–2.4 | <0.001 |
| **Parents Smoking Status (ref: NO)** | 1.3 | 1.0–1.6 | <0.001 | 0.6 | 0.5–0.8 | <0.001 |
| **Tobacco logos/print (ref: NO)** | 3.4 | 3.0–3.9 | <0.001 | 1.4 | 1.1–1.8 | 0.009 |
| **Event Promotion (ref: NO)** | 1.7 | 1.4–2.1 | <0.001 | 1.2 | 0.9–1.6 | 0.3 |
| **Student Type** | | | | | | |
| Boarding Student | Ref | - | - | Ref | - | - |
| Day Scholar Student | 0.9 | 0.7–1.2 | 0.63 | 1.3 | 1.0–1.7 | 0.045 |
| **Age of the Survey Respondents** | | | | | | |
| 12 years or younger | Ref | - | - | Ref | - | - |
| 13–15 years old | 4.70 | 2.01–11.0 | <0.001 | 2.64 | 1.17–6.65 | 0.027 |
| 16 years or older | 11.3 | 5.10–24.9 | <0.001 | 3.34 | 1.44–8.63 | 0.008 |
| **Grade of the student** | | | | | | |
| High Schoolers | Ref | | - | Ref | | - |
| Junior High Schoolers | 0.5 | 0.4–0.6 | <0.001 | 1.4 | 1.0–1.8 | 0.035 |
| **See teachers Smoking (ref: NO)** | 2.6 | 2.1–3.2 | <0.001 | 1.4 | 1.0–1.9 | 0.042 |
| **Media Promotion (ref: NO)** | 1.6 | 1.3–2.0 | <0.001 | 1.3 | 1.0–1.7 | 0.075 |
| **Social Comfort with Smoking** | | | | | | |
| No Difference | Ref | - | - | | | |
| More Comfortable | 1.9 | 1.5–2.4 | <0.001 | - | - | - |
| Less Comfortable | 1.2 | 1.0–1.4 | 0.08 | - | - | - |
| **Think it it's difficult to quit (ref: NO)** | 0.9 | 0.7–1.1 | 0.25 | - | | - |
| **Danger of Tobacco use (ref: Yes)** | 1.1 | 0.5–0.9 | 0.62 | - | | - |
| **Tobacco Use seen on Screen (ref: NO)** | 1.3 | 1.0–1.7 | 0.11 | - | | - |
| **Sex of the respondents (ref: Male)** | 0.3 | 0.2–0.3 | <0.001 | - | | - |
| **Pocket Money** | | | | | | |
| No Pocket Money | Ref | - | - | | | |
| Less than Nu. 500 | 1.4 | 1.1–1.8 | <0.001 | - | | - |
| More than Nu. 500 | 2.7 | 1.9–3.7 | <0.001 | - | | - |

"having tried other smoking products", "exposed to secondhand smoking", "cost of the tobacco", "best friends smoking status", "parents smoking status", "seeing teachers smoke", "grade", "having tobacco print accessories", "event and media promotion", and "student type" as key correlates for the adolescent tobacco use in Bhutan.

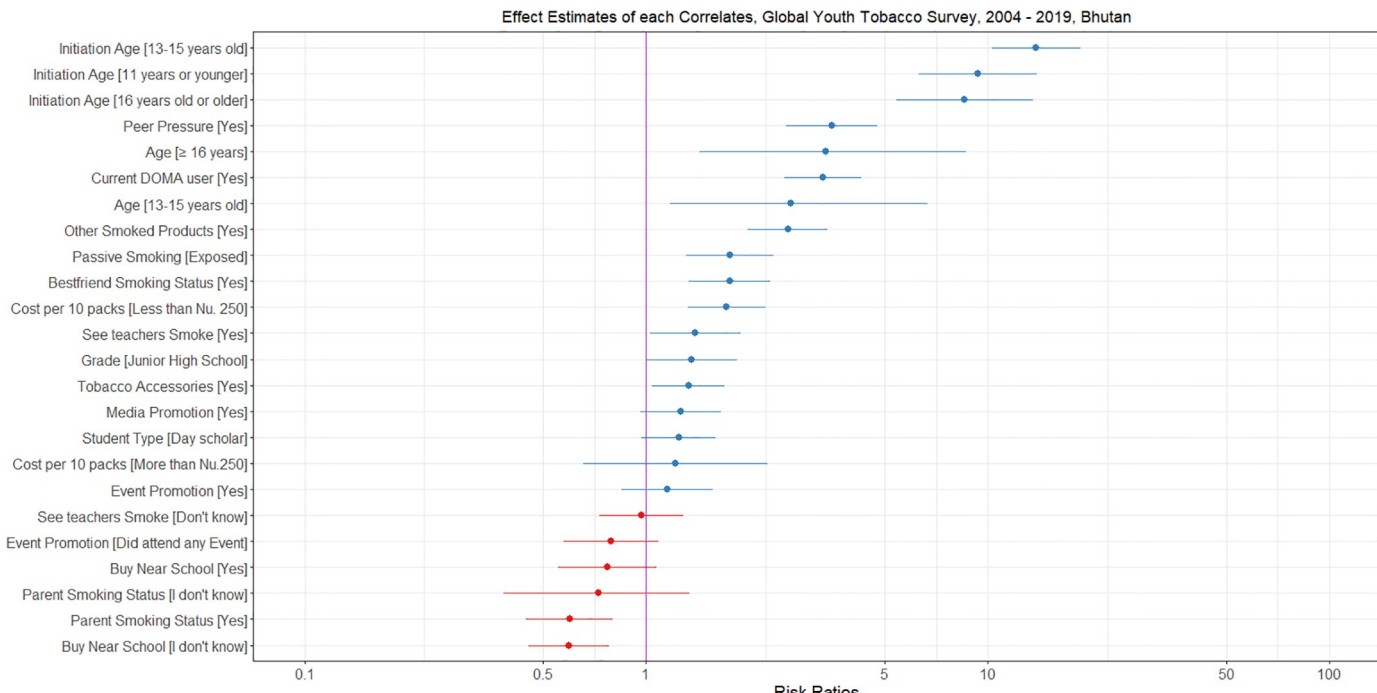

**Fig 2. Effect Estimates of each correlates, Global Youth Tobacco Survey, 20024–2019, Bhutan.**

The age of initiation to tobacco use was strongly associated with higher likelihood of tobacco use. Age of first tobacco exposure before 11 years and between 13–16 years had higher odds of becoming future tobacco users (aOR 9.2, 95% CI 6.2–13.7, p<0.001) and (aOR 12.8, 95% CI 9.6–17.1, p<0.001), respectively compared to those who were never tried smoking. Respondents who were "current DOMA users" had 3.3 times the odds of using tobacco (95% CI 2.5–4.2, p<0.001) compared to those who were not "current DOMA users". Respondents who had peer pressure had 3.6 times higher odds (95% CI 2.6–4.8, p<0.001) of tobacco use compared to those who did not have any peer pressure from their friends. Students who used other forms of smoking products had 2.7 (95% CI 2.1–3.5, p<0.001) times higher odds of using tobacco compared to those who did not use them. The cost of the cigarette was also shown to be strong predictor for current tobacco use. Fig 2, shows the effect estimates of each correlate.

Exposure to second hand smoking at home showed significant association with tobacco use (aOR 1.8, 95% CI 1.4–2.4, p<0.001). However, parents smoking status was associated with lower likelihood of tobacco in students (aOR 0.6, 95% CI 0.5–0.8, p<0.001). "Owning items with tobacco logos" increased odds of tobacco use (aOR 1.4, 95% CI 1.1–1.8, p = 0.009). Students who were day scholars had higher odds of using tobacco compared to the boarding students (aOR 1.3, 1.0–1.7, p = 0.045). Older aged students had higher odds of using tobacco (13–16 years old aOR 2.8 and ≥17 years old 3.4 respectively) compared to ≤12-year-olds. Junior high schoolers were more likely to use tobacco compared to high schoolers (aOR 1.4, 1.0–1.8, p = 0.035). Same was true for students who saw teachers smoking in the schools (aOR 1.4, 1.0–1.9, p = 0.042).

## Discussion

Trends in prevalence of tobacco use between 2004–2019 showed alarming rise in tobacco use among students. This is more markedly seen in male respondents. While smoking prevalence

has increased gradually, smokeless tobacco use increased more rapidly in both sexes over the recent years. This highlights an important trend with smokeless tobacco emerging as a major form of adolescent tobacco use. Our findings suggest age of initiation of tobacco use, current betel quid users, peer pressure, and price of tobacco products as major correlates of adolescent tobacco use.

Reviews across neighboring countries show that Bhutan has highest prevalence of youth tobacco use in the region. A recent demographic and health survey in Nepal showed a 30% prevalence of tobacco use among adults above 15 years of age [12]. The GYTS study conducted in Nepal in 2011 reported a prevalence of 20.4% a while Global School-Based Student Health Survey conducted in 2015 observed a lower overall tobacco use prevalence of 9% among the youth population [13, 14]. In contrast to Bhutan, Bangladesh reported a decreasing trend of tobacco based on two consecutive survey waves' data [15, 16]. India, characterized by diverse demographic heterogeneity, reported a substantial availability of tobacco products in the market with male students, exhibiting higher susceptibility to tobacco consumption [17]. The GYTS study conducted in India during 2016–2017 reported an overall prevalence of 11.9% tobacco use among Indian adolescents [18]. Thus, Bhutan seems to have highest prevalence of student tobacco consumption in the region. As a remedial measure, continued surveillance of tobacco control efforts should be emphasized continuously in Bhutan.

Age of initiation of tobacco use before 16 years of age, student ages between 13–17 years old, current betel quid users, students using other forms of smoked products, second-hand smoking and best friends smoking status were observed as significant correlates of tobacco use. These findings suggests that junior high schoolers in Bhutan are at most risk of indulging in tobacco use. In contrast, an examination of GYTS data from chosen African countries did not observe age as a significant predictor while other factors such as secondhand smoking and disposable income emerged as significant predictors of adolescent tobacco use [19]. Similar findings were also reported from neighboring Nepal with age and peer smoking status being identified as correlates based on a cross sectional survey [20].

The use of betel quid locally called DOMA, is pervasive in Bhutanese society [21] and identified as a predictor of tobacco use in this study. This is consistent with earlier reports from Taiwan showing combined effect of betel quid and tobacco use [22]. Similarly, a study conducted in Myanmar also reported high prevalence of the dual use of tobacco and betel quid [23]. Peer pressures, such as "willingness to smoke if best friend offers", and "having a friend who is smoker", parental smoking status and secondhand smoking at home were also identified as predictors of tobacco use. Another study from Nepal showed similar association based on factors related to peers and family influence dynamics [24]. Similar findings were also reported from analysis of five central and eastern European countries on youth tobacco use, which are consistent with our findings [25]. Tobacco costs and disposable income identified as predictors of adolescent tobacco use in this study are also consistent with a published study [26].

Since the adoption of WHO Convention Framework for Tobacco Control (WHO-FCTC), Bhutan had adopted, amended, and changed number of rules and regulations regarding tobacco use in the country. These include the Tobacco Control Act of Bhutan 2010 [27], Tobacco Control (Amendment) Act of Bhutan 2012 [28], Tobacco Control (Amendment) Act of Bhutan 2014 [6], and Tobacco Control (Amendment) Act 2021 [29]. Between 2009 and 2013, numerous legislations regarding tobacco use have been implemented in Bhutan. The Tobacco Control Act of 2010 instituted a comprehensive set of tobacco control measures. Cultivation of tobacco, manufacturing, distributing, promoting, and selling of tobacco products were all deemed illegal [27]. It also mandated smoke-free public areas, required health warnings and country-of-origin labels on tobacco packaging, and enabled hefty import taxes [30].

This further cemented with Tobacco Control Regulations, 2013 and Tobacco Control Amendment Act of 2014. Despite this stringent rules, the tobacco was accessible to Bhutanese students through constant smuggling and black market [31]. Additionally, during the time of COVID-19, the government relaxed the rules on tobacco control measures.

The study highlights the imperative need for targeted interventions aimed at prevention and mitigation strategies specifically designed to address the prevalence of tobacco use among students. A comprehensive understanding of the determinants behind the increasing trend for tobacco use is essential for developing effective public health initiatives. Bhutanese society sees tobacco use negatively in both cultural and religious viewpoints while use of betel quid is a culturally accepted phenomenon. Findings from this study suggest the imperative need to accommodate use of both tobacco and betel quid in the intervention strategies among students.

The findings from this study should be interpreted taking into consideration the limitations of the study. First, we used cross-sectional Global Youth Tobacco Survey data, so causation between the identified predictors and tobacco use cannot be established. The GYTS study is based on self-reported survey data, which may have been affected by recall bias. Additionally, the GYTS standard protocol was updated in 2012 to ensure tobacco product use definitions across countries [32]. Changes in question wording may have affected student responses in subsequent surveys. Furthermore, the findings on prevalence change might be influenced by the varying intervals at which Bhutan conducted the GYTS survey—two, three, four, and six years, respectively. However, the study findings on adolescent tobacco use determinants and its prevalence over the survey years can help guide the future tobacco policies in the country.

## Conclusion

The prevalence of tobacco uses among school students in Bhutan is one of the highest in the region. Male students had higher prevalence compared to females with annual average percentage change of 2.4%. Smokeless tobacco use showed a two-fold annual average percent change (5.8%) compared to smokers. Age of exposure, current betel quid users, peer pressure, and price of tobacco products are shown to be associated with adolescent tobacco use in Bhutan. Concerted governmental, and multi-stakeholder solutions need to be implemented across all the schools in Bhutan through aggressive advocacy and awareness programs. Junior high school age students maybe targeted in limited financial resource settings.

## Supporting information

**S1 Data.**
(PDF)

## Acknowledgments

We would like to thank the Ministry of Health, Royal Government of Bhutan for conducting the primary surveys and making the data available for public use. We also would like to thank the WHO country office in Bhutan and the Center for Disease Control, USA for funding the primary surveys.

## Author Contributions

**Conceptualization:** Tshewang Gyeltshen, Mahbubur Rahman.

**Data curation:** Tshewang Gyeltshen.

**Formal analysis:** Tshewang Gyeltshen.

**Investigation:** Tshewang Gyeltshen.

**Methodology:** Tshewang Gyeltshen, Mahbubur Rahman.

**Project administration:** Tshewang Gyeltshen, Mahbubur Rahman.

**Resources:** Tshewang Gyeltshen.

**Supervision:** Mahbubur Rahman.

**Validation:** Tshewang Gyeltshen.

**Visualization:** Tshewang Gyeltshen.

**Writing – original draft:** Tshewang Gyeltshen.

**Writing – review & editing:** Tshewang Gyeltshen, Mahbubur Rahman.

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
