## [Decision Letter · Decision Letter 0]

27 May 2024

PGPH-D-24-00020

Trends in prevalence and correlates of tobacco use among school-going adolescents in Bhutan: A secondary data analysis of the 2004-2019 Global Youth Tobacco Survey.

Dear Dr. GYELTSHEN,

Thank you for submitting your manuscript to PLOS Global Public Health. After careful consideration, we feel that it has merit but does not fully meet PLOS Global Public Health’s publication criteria as it currently stands. Therefore, we invite you to submit a revised version of the manuscript that addresses the points raised during the review process.

We look forward to receiving your revised manuscript.

Kind regards,

Biplab Datta, Ph.D.

Academic Editor

Journal Requirements:

1. Please amend your online Financial Disclosure statement. If you did not receive any funding for this study, please simply state: “The authors received no specific funding for this work.”

2. Please update your online Competing Interests statement. If you have no competing interests to declare, please state: “The authors have declared that no competing interests exist.”

3. Please provide separate figure files in .tif or .eps format only and remove any figures embedded in your manuscript file. Please also ensure that all files are under our size limit of 10MB. You may leave the figure captions or legends in the manuscript.

Additional Editor Comments (if provided):

Reviewers' comments:

Reviewer's Responses to Questions

**Comments to the Author**

1. Does this manuscript meet PLOS Global Public Health’s publication criteria? Is the manuscript technically sound, and do the data support the conclusions? The manuscript must describe methodologically and ethically rigorous research with conclusions that are appropriately drawn based on the data presented.

Reviewer #1: Yes

Reviewer #2: Partly

2. Has the statistical analysis been performed appropriately and rigorously?

Reviewer #1: Yes

Reviewer #2: Yes

3. Have the authors made all data underlying the findings in their manuscript fully available (please refer to the Data Availability Statement at the start of the manuscript PDF file)?

Reviewer #1: Yes

Reviewer #2: Yes

4. Is the manuscript presented in an intelligible fashion and written in standard English?

Reviewer #1: Yes

Reviewer #2: Yes

5. Review Comments to the Author

Reviewer #1: This is a very important research work. However, the authors should take note of the following.

General comments

GYTS is designed to be representative of youths 13-15 years, but the authors did not specify why the decided to focus on a different age cohort. Moreover, it is difficult for readers to know which age cohort was used as there is no consistency in the use of the age cohort included in the analysis. Sometimes you use 11-17, 11-18 and 11-19 years and line 22 indicates under 11 years, what was the minimum and maximum age?

Why was the term adolescent used instead of youth since you are using GYTS data? I understand adolescent covers the ages included in your study, but it is better to be consistent with the GYTS study protocol.

Please take note of the spelling and grammatical errors. Consider getting the services of a native English speaker to proofread the document.

Specific comments

Introduction

Line 45- 49 -Do you mean tobacco use is was banned? Did enactment of the Tobacco Control Act and enforcement of the ban occur at the same time? The sequence is not clear and might be confusing to readers.

Line 53-55, report the actual prevalence and mention the year.

Methods

Line 62 – Mention the total number of time points and survey years.

Line 77-78- Revise the sentences.

Line 83-85- Reframe this sentence “The outcome variable for the study was tobacco use- defined as use of any form of tobacco (smoked/smokeless) currently using at the time of the survey.”

Can you please define the key variables used in your analysis?

Specify if all the surveys included in your analysis were national or subnational surveys.

GYTS uses a complex survey design approach, was this controlled for in the analysis? I understand you mentioned in lines 109-110 that a weighted analysis was conducted but you did not specify what the weighting controlled for. Why did you not report standard errors and/or confidence intervals to enable readers assess the precision of your point estimates?

Results

Lines 143 to 144 Revise this sentence: “The prevalence of tobacco uses “ever used".

tobacco”, “current use”, “smoked”, and “smokeless” are presented separately.”

Lines 158-159: The sentence is difficult to understand, consider revising it

Lines 177 to 179 -How was first tobacco exposure before 11 years defined.

Line 190 the GYTS suggestion to use the term “exposure to secondhand smoke” instead of “passive smoking” as this is the term used in the GYTS tool.

Discussion

Lines 201-203 (first and second sentences) do not fit here and should be deleted.

Reviewer #2: General comments:

For the sake of clarity and precision, I strongly recommend using the term 'students' instead of 'youth' in your manuscript. This is particularly important as the GYTS is a nationally representative survey of students in public and private schools, and it does not encompass all youth, especially those not educated in formal settings.

The methods section did not include the measures /definition used for smoked and smokeless tobacco. At a minimum, definitions for current cigarette smoking, current other smoked tobacco products, and current smokeless tobacco use should be provided.

Given that GYTS targets students aged 13-15 and the chosen analytic sample is heavily weighted toward this population group, I suggest focusing your analysis on this age range. Not doing so may bias your findings toward this age group.

Current and ever use are not defined in the manuscript but are used throughout.

The manuscript needs editorial review to address minor grammatical issues throughout the paper.

Introduction

Line 32: Suggest using tobacco use instead of “uses.”

Line 33: Suggest saying “nicotine addiction” or “nicotine dependence” instead of addiction.

Lines 33-35: Suggest using the latest estimate included in the most recent WHO global trends report (WHO global report on trends in prevalence of tobacco use 2000–2030. Geneva: World Health Organization; 2024.). For line 35, I recommend using the estimate for tobacco use, not smoking, given that your manuscript is focused on tobacco use.

Line 36: Please add a comma after “world.”

Line 37: Please capitalize the first letters in the global youth tobacco survey.

Line 44: Spell out WHO since it's your first time using the initials.

Line 51: The GYTS is part of the Global Tobacco Surveillance System, which includes multiple partners, such as the US Centers for Disease Control, which provides technical assistance. Given the partnership, I suggest not singling out WHO here.

Methods

Line 67-68: Please add a citation for the following statement: “Bhutan is also known internationally for its measure of 68 Gross National Happiness as a development Index.”

Line 72-75: The authors should include a justification for why their analytic sample included students aged 11-18 when GYTS is designed to target students aged 13-15.

Line 76-79: This paragraph can be deleted as it does not add much to the manuscript given that the public was not involved.

Line 81: Suggest removing “WHO” per my comment regarding line 51.

Line 83-85: “Current use” in GYTS is defined as use within the past 30 days, not necessarily use at the time of the survey. Please revise this statement.

Line 100-101: Please provide the citation for this website.

Line 106-108: Please indicate which R package was used to conduct the analysis, if any.

Line 199: Please indicate what you will be adjusting/controlling for in your model.

Results

Line 130-131: The methods section did not describe exposure to secondhand smoke as an outcome. Why is it being reported in the results? Clarifying all measures and their definition in the methods section would be good.

Line 140: given the target age from GYTS is 13-15, I suggest recognizing the age groups as ≤12 years; 13-15; ≥16 years

Line 156: The authors used sex throughout the paper but proceeded to use gender here. Please be consistent with sex and gender, as they are not interchangeable.

Line 159-160: The authors mentioned that smokeless tobacco use increased two-fold (AAPC) compared with smokers throughout the survey years. This statement is misleading without knowing the percentage for the latter. Please include the percentage in parentheses.

Line 161: Figure 1 is not easy to understand as it is difficult to understand the provided. Are the solid lines males and dotted line females? I suggest recreating the graphic to make it easier for readers to understand.

Discussion

Lines 262-267: A significant study limitation is that the initial GYTS core questionnaire from 1999 was updated in 2012. The wording for some questions may have changed, likely affecting how students responded in subsequent surveys. Also, the findings for prevalence change might be influenced by the periodicity in which Bhutan conducted the GYTS survey, as the interval between rounds was two years, three years, four years, and six years, respectively.

6. PLOS authors have the option to publish the peer review history of their article (what does this mean?). If published, this will include your full peer review and any attached files.

**Do you want your identity to be public for this peer review?** For information about this choice, including consent withdrawal, please see our Privacy Policy.

Reviewer #1: No

Reviewer #2: No

---

## [Editor Report · Decision Letter 1]

9 Jul 2024

Trends in prevalence and correlates of tobacco use among school-going adolescents in Bhutan: A secondary data analysis of the 2004-2019 Global Youth Tobacco Survey.

PGPH-D-24-00020R1

Dear Dr. GYELTSHEN,

We are pleased to inform you that your manuscript 'Trends in prevalence and correlates of tobacco use among school-going adolescents in Bhutan: A secondary data analysis of the 2004-2019 Global Youth Tobacco Survey.' has been provisionally accepted for publication in PLOS Global Public Health.

Best regards,

Biplab Datta, Ph.D.

Academic Editor